# Effects of standard and low doses of estradiol on markers of endometrial receptivity in frozen-thawed embryo transfer cycles: Double-blind, randomized clinical trial

Nichamon Parkpinyo[1], Isarin Thanaboonyawat[1], Japarath Prechapanich[1], Pitak Laokirkkiat[1], Roungsin Choavaratana[1], Suchanan Hanamornroongruang[2], Somsin Petyim[1,3]*

1 Reproductive Biology and Infertility Unit, Department of Obstetrics and Gynecology, Faculty of Medicine Siriraj Hospital, Mahidol University, Bangkok, Thailand, 2 Department of Pathology, Faculty of Medicine Siriraj Hospital, Mahidol University, Bangkok, Thailand, 3 Center of Reproductive and Stem Cell Biology Unit, Department of Obstetrics and Gynecology y, Faculty of Medicine Siriraj Hospital, Mahidol University, Bangkok, Thailand

* somsin101@yahoo.com

## Abstract

### Background

Endometrial receptivity biomarkers, specifically the regulatory proteins HOXA-10 and HOXA-11 as well as the integrin αvβ3, play vital roles in implantation during the window of implantation. This Double-blinded, randomized clinical trial compares the effects of two initial doses of oral estrogen regimens on endometrial receptivity markers during the window of implantation in endometrial preparation for frozen-thawed embryo transfer.

### Methods and findings

The study includes infertile patients who underwent in vitro fertilization and planned frozen-thawed embryo transfer at the infertility clinic between June 2018 and March 2019. Fifty patients were randomized to a low-dose (4 mg/day) or standard-dose (6 mg/day) estradiol group for artificial endometrial preparation. On the first visit (day 12 of the cycle), measurements of mean endometrial thickness and estradiol and progesterone levels were taken. Following this visit, patients received 600 mg daily of micronized progesterone for 7 days. On the second visit (day 19 of the cycle), hormonal levels were reassessed, and an endometrial biopsy was performed for immunohistochemical analysis. The primary outcome was the expression level of HOXA-10. Secondary outcomes included the expression levels of HOXA-11 and integrin αvβ3, mean endometrial thickness, as well as serum estradiol and progesterone levels measured at various phases of the endometrial cycle. The mean age of the

**Data availability statement:** All relevant data are within the paper and its Supporting Information files.

**Funding:** We received Siriraj Grant for Research Development, Faculty of Medicine Siriraj Hospital, Mahidol University (grant number R016133010)]; however, the funders had no role in study design, data collection and analysis, the decision to publish, or preparation of the manuscript.

**Competing interests:** The authors have declared that no competing interests exist.

participants was 36 years. The standard-dose group exhibited significantly greater intensity scores for the expression of the regulatory proteins HOXA-10 and HOXA-11 and the integrin αVβ3 than did the low-dose group ($75.84 \pm 11.25$ vs $61.53 \pm 11.05$, $107.08 \pm 19.42$ vs $87.62 \pm 9.40$, and $90.25 \pm 10.42$ vs $76.32 \pm 12.98$, respectively; $P < 0.001$). The groups had no significant differences in mean serum estradiol level, progesterone level, or endometrial thickness during the artificial cycle at the first or second visit.

## Conclusions

Optimal artificial endometrial preparation for embryo transfer in a frozen-thawed embryo transfer cycle is crucial for maximizing implantation outcomes. This study suggests that the administration of a fixed standard dose of 6 mg of estradiol for artificial endometrial preparation should be considered.

## Introduction

Globally, in vitro fertilization is being increasingly utilized. Advances in assisted reproductive technologies and ovarian hyperstimulation protocols have enabled the cryopreservation of standard-quality embryos for subsequent frozen-thawed embryo transfer cycles. Understanding the nuances of endometrial preparation in the frozen-thawed embryo transfer cycle is pivotal for successful implantation [1]. Exogenous administration of estrogens and progestogens in a precise sequence can artificially induce optimal endometrial preparation [2]. Various protocols for artificial endometrial preparation have been explored, with constant and increasing doses of estrogen being the most noteworthy. Madero et al reported a significantly greater biochemical pregnancy rate with a constant-dose regimen than with an increasing-dose protocol [3].

Embryo implantation, a critical determinant of pregnancy success, progresses through three stages: apposition, adhesion, and invasion [4]. Success hinges on synchronizing embryo development and endometrial receptivity [5]. While impaired embryogenesis accounts for approximately one-third of implantation failures, aberrant endometrial receptivity is responsible for the remaining cases [6]. Endometrial receptivity, particularly during the window of implantation, is instrumental in accepting an embryo. The window of implantation is a finite period characterized by critical cellular and molecular events that facilitate implantation, typically occurring between days 20 and 24 of the menstrual cycle or 7–9 days postovulation [7,8].

Diagnostic methods for determining endometrial receptivity include endometrial thickness measurement, endometrial histological dating [9], and molecular marker assessment via transcriptomics [10,11], proteomics [12], and secretomics [13]. Biomarkers, specifically the regulatory proteins HOXA-10 and HOXA-11 as well as the integrin αvβ3, play vital roles in implantation during the window of implantation [14–18]. The HOXA-10 and HOXA-11 transcriptional regulators are significantly expressed in the endometrial glands and stroma during the mid-luteal phase, the time

when implantation occurs [17]. These regulators are vital for uterine organogenesis and functional differentiation of the endometrium [15]. Integrins are essential heterodimeric glycoproteins that fulfill numerous roles within the endometrium [6]. Notably, the integrin αvβ3, which is crucial for implantation, appears on luminal and glandular cell surfaces as the window of implantation commences.

Previous studies examining estrogen use in artificial endometrial preparation for frozen-thawed embryo transfer cycles have focused primarily on endometrial thickness or histological dating, with neither providing a full representation of endometrial quality or quantity [19]. The influence of endometrial factors on pregnancy outcomes presents challenges in accurately assessing endometrial receptivity [3,20]. Optimal estradiol dosing is crucial for developing a standard-quality endometrium, creating the conditions for a progesterone-induced secretory phase and ensuring a receptive endometrium for embryo implantation.

This double-blinded, randomized clinical trial aimed to evaluate the impact of two different estrogen doses on endometrial receptivity markers during the window of implantation in endometrial preparation for frozen-thawed embryo transfer. The primary outcome was the expression of the endometrial receptivity gene *HOXA-10*. Secondary outcomes included the expression of other endometrial receptivity markers—*HOXA-11* and integrin αvβ3—along with mean endometrial thickness, and serum levels of estradiol and progesterone measured at various stages of the endometrial cycle.

## Methods

### Study design

This double-blinded, randomized clinical trial was conducted over 10 months from June 2018 to March 2019 at the infertility clinic of the Faculty of Medicine Siriraj Hospital, Bangkok, Thailand. The Siriraj Institutional Review Board authorized the study protocol (approval number 653/2560[EC2]), and the research was registered with the Thai Clinical Trials Registry (TCTR20180718003). Before study enrollment, all participants received comprehensive counseling and provided written informed consent.

### Study population

Fifty infertility patients scheduled for frozen-thawed embryo transfer in the in vitro fertilization program were recruited for the study. The inclusion criteria were age between 18 and 45 years; normal follicle-stimulating hormone, luteinizing hormone, and estradiol levels during the early follicular phase; and regular menstrual cycles during the 3 months before enrollment. The exclusion criteria were contraindications to estrogen use (for example, endometrial cancer, unexplained bleeding, thromboembolism, or liver disease); uterine cavity abnormalities (such as synechiae or polyps); or estrogen use during the 3 months before enrollment.

### Study procedure

The demographic data collected comprised age, body mass index, live birth and miscarriage history, curettage history, and infertility duration. Patients were randomized into low-dose or standard-dose estradiol hemihydrate (Estrofem) groups using computer-generated randomization, with 25 individuals per group. Allocation concealment was maintained with sealed envelopes, and both patients and the investigator were blinded to the assignments. The gynecologists who assessed the endometrial thickness were also blinded to the treatment allocation. The low-dose group received 4 mg/day of EstroFem, which was taken as two capsules containing 1-mg of Estrofem in the morning and evening (total dose is 4 mg/day). The standard-dose group was prescribed 6 mg/day, which involved taking one capsule containing 1-mg of Estrofem and one capsule containing 2-mg of Estrofem twice daily (total dose is 6 mg/day). The capsules containing 1-mg and 2-mg of Estrofem look alike. Treatment began on day 2 of the menstrual cycle and continued for 10 days.

 

At the initial follow-up, which was conducted on day 12 of the menstrual cycle, endometrial thickness was assessed via transvaginal ultrasonography (Xario 100; Toshiba Medical Systems, Japan) via a sagittal view of the uterus. A single experienced gynecologist, who was blinded to the treatment allocation, performed this assessment. Blood samples for estradiol and progesterone level determination were drawn. The compliance with the medication was verified during the study period by counting the returned capsules in the medication pack. Following the initial visit, participants commenced a regimen of micronized progesterone (Utrogestan), prescribed at a daily dosage of 600 mg. This dosage was administered in three doses daily for 7 consecutive days. At the second follow-up (day 19 of the cycle), blood samples were collected to measure the estradiol and progesterone levels, and an endometrial biopsy was obtained. In the subsequent menstrual cycle, standard procedures for endometrial preparation were followed in anticipation of frozen-thawed embryo transfer.

## Endometrial biopsy procedure

During the second follow-up (day 19 of the menstrual cycle), an endometrial biopsy was performed on all patients utilizing the endometrial aspiration technique with a Wallach Endocell endometrial sampling device (CooperSurgical, USA). The procedure entailed the insertion of a vaginal speculum, followed by the identification and cleaning of the cervix and cervical canal. The cervical mucus was removed, and the cervix was disinfected with a normal saline solution. The Endocell device was then carefully inserted through the cervical os to the uterine fundus, where negative pressure was applied to aspirate the endometrial tissue. Sampling occurred continuously throughout the uterine cavity, and each procedure was carried out by the same operator. The collected endometrial tissue was promptly placed in a tissue cassette for paraffin embedding and submerged in 10% neutral-buffered formaldehyde for fixation overnight in preparation for subsequent immunohistochemical analysis. Morphological examination of all endometrial tissues was performed according to the histological dating criteria established by Noyes [21]. To ensure objectivity, a senior pathologist, who was blinded to the study details, was tasked with interpreting the histological dating and immunobiological findings.

## Immunohistochemical staining protocol

After endometrial biopsy, the specimens were immediately fixed in 10% neutral-buffered formaldehyde and subsequently embedded in paraffin wax. The embedded tissues were sectioned into 3-μm slices, affixed onto SuperFrosted Plus adhesion slides (Thermo Scientific, UK) and incubated overnight at 60°C. The specimens were then autostained using the Ventana BenchMark Ultra Staining System (N750-BMKU-FS 05342716001; Roche) following an optimized immunohistochemical staining protocol for detecting the expression of the regulatory proteins HOXA-10 and HOXA-11 and the integrin αvβ3. For HOXA-10 detection, antigen retrieval was performed at 95°C for 64 minutes in OptiView CC1 (pH 6.0), and sections were then incubated with a rabbit polyclonal anti-human HOXA-10 primary antibody (GTX37412, GeneTex) at a 1:50 dilution at 37°C. The milder protocol for HOXA-11 and integrin αvβ3 involved antigen retrieval in Mild Ultra CC1 (pH 6.0) for 52 minutes. This was followed by incubation with rabbit polyclonal anti-human HOXA-11 (GTX48983, GeneTex, 1:200 dilution) and mouse monoclonal anti-human αVβ3 integrin (MAB1976, Merck, 1:1500 dilution) primary antibodies at 37°C. Detection was carried out using an Ultraview Universal DAB Detection Kit. Finally, the sections were counterstained with hematoxylin, mounted with Permount solution, and secured with a coverslip. The process of staining and analysis of the specimens for the HOXA-10, HOXA-11 and for integrin αvβ3 expression were standardized before assessment.

Control measures included the use of slides of normal secretory-phase endometrial epithelium with referral staining procedures as positive controls. Slides processed without primary antibodies served as negative controls.

## Immunohistochemical staining analysis

The intensity of immunohistochemical staining for the regulatory proteins HOXA-10 and HOXA-11 and for integrin αvβ3 was quantified via intensity scores using ImageJ 1.46r/Java 1.6.0_20 (64-bit, USA). This scoring focused on the glandular

epithelium of each section and was independently conducted by two observers. The final scores for each protein were derived by averaging the scores assigned to each compartment.

## Statistical analysis

Demographic and descriptive statistics are presented as the mean ± SD or median (min–max), as appropriate. Comparative analyses between groups were conducted using an independent $t$ test for normally distributed data and the Mann–Whitney U test for nonnormally distributed data. Pearson's correlation was used to assess the relationship between the mean endometrial thickness and immunohistochemical staining intensity in the different groups. Data analysis was performed using PASW Statistics, version 18 (SPSS Inc, Chicago, IL, USA), with $P$ values < 0.05 indicating statistical significance.

The sample size calculation was based on the previous study [17]. The sample size was calculated based on HOXA-10. HOXA-10 functions as an upstream master regulator, controlling the expression of other key genes involved in uterine development and endometrial receptivity [17]. The formula for testing two independent means was used. With a Type I error (α) of 0.05 and a power of 90%, a sample size of 46 women was necessary for the study. When accounting for a dropout rate of 10%, the total sample size needed was 50 women. The appropriate statistical test is the Independent samples t-test. The effect size measured using Cohen's d is 0.97.

## Results

Among the 78 patients assessed for eligibility, 50 were included in this study and randomized into two groups. In the low-dose group, one patient's data were excluded due to a diagnosis of endometrioid adenocarcinoma of the uterus identified in the final pathological report. In the standard-dose group, data from one patient were discarded due to incorrect drug consumption. Therefore, the analyses were performed on a per-protocol basis (Fig 1). All 48 patients in the final analysis compliant to study protocol. Demographic and clinical characteristics (age, body mass index, number of live births and miscarriages, history of curettage, and duration of infertility) were recorded (Table 1). The mean ages were 36.25 ± 4.28 years in the low-dose group and 36.92 ± 2.60 years in the standard-dose group.

### Expression of the regulatory proteins HOXA-10 and HOXA-11 and integrin αvβ3

Immunohistochemical analysis revealed differential expression of the regulatory proteins HOXA-10 and HOXA-11 and of integrin αvβ3 between the groups. The intensity of HOXA-10 expression was significantly greater in the standard-dose group than in the low-dose group (75.84 ± 11.25 vs 61.53 ± 11.05; $P$ < 0.001; Figs 2, 3a, and 3b). Similarly, HOXA-11 expression was significantly greater in the standard-dose group (107.08 ± 19.42 vs 87.62 ± 9.40; $P$ < 0.001; Figs 2, 3e, and 3f). The intensity of integrin αvβ3 expression was also significantly greater in the standard-dose group (90.25 ± 10.42 vs 76.32 ± 12.98; $P$ < 0.001; Fig 2, 3i, and 3j).

### Localization of the regulatory proteins HOXA-10 and HOXA-11 and integrin αvβ3

The specificity of the immunohistochemical staining was confirmed using normal secretory-phase endometrial epithelium as a positive control. HOXA-10 was predominantly localized in the cytoplasm of glandular epithelial cells and was less abundant in the nuclei of stromal cells (Fig 3c). HOXA-11 exhibited similar cytoplasmic localization (Fig 3g). The integrin αVβ3 was expressed in both glandular epithelial and stromal cells (Fig 3k). Histological evaluation of endometrial biopsies confirmed an adequate secretory-phase endometrium for days 18 and 19, according to the Noyes criteria, characterized by decreased vacuolization, basal nuclei, and improved cellular alignment.

### Hormone levels and endometrial thickness

The serum hormone levels and mean endometrial thickness data collected during the artificial cycle are detailed in Table 2. The mean serum estradiol levels at the first visit were 220.55 ± 117.25 pg/ml in the low-dose group and 229.98 ± 80.06 pg/

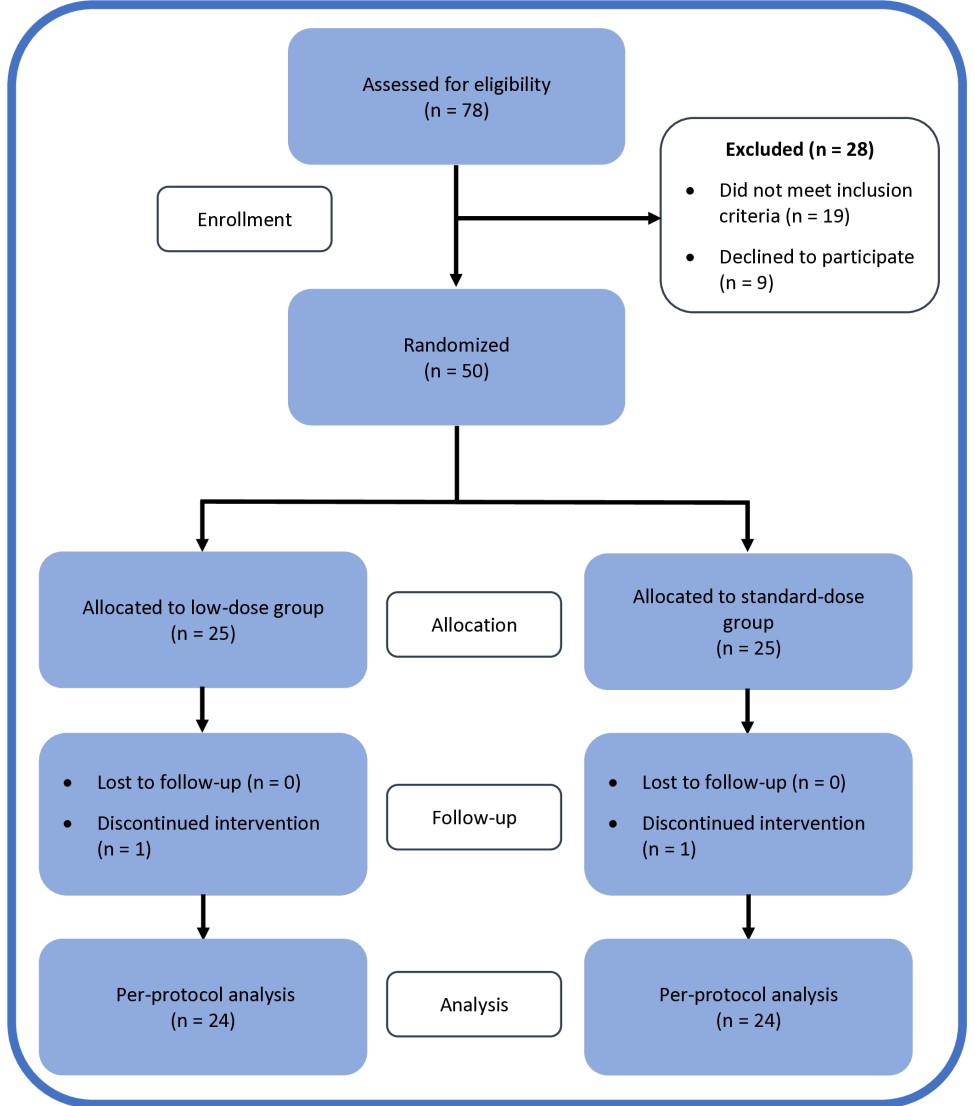

**Fig 1. Flow diagram of the double-blinded, randomized clinical study design.**

**Table 1. Demographic and clinical characteristics and outcomes of the study participants.**

| Variables | Low-dose estradiol (n=24) | Standard-dose estradiol (n=24) | P |
|---|---|---|---|
| Age (yrs)[a] | 36.25±4.28 | 36.92±2.60 | 0.517 |
| Body mass index (kg/m²)[a] | 22.40±4.33 | 22.79±4.76 | 0.775 |
| Duration of infertility (yrs)[a] | 6.79±3.68 | 6.71±3.21 | 0.934 |
| Live births[b] | 0 (0-3) | 0 (0-1) | 0.292 |
| Miscarriages[b] | 0 (0-3) | 0.5 (0-3) | 0.313 |
| Curettages[b] | 0 (0-2) | 0 (0-2) | 0.196 |

Note:[a]The data are presented as means±standard deviations.

[b]The data are presented as medians (min–max).

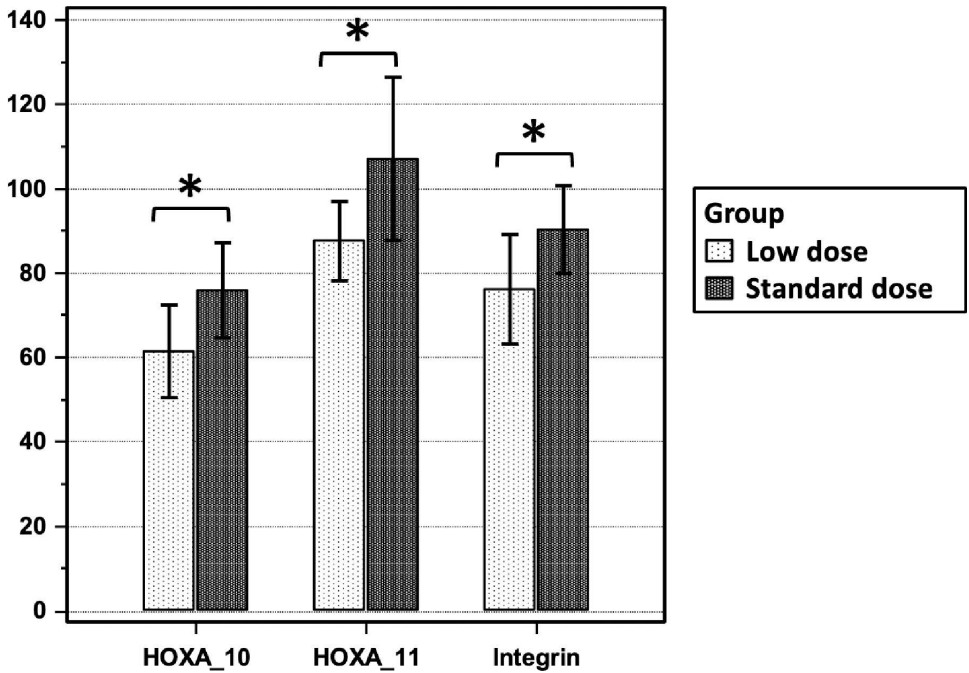

**Fig 2. Comparative immunohistochemical staining intensities of the regulatory proteins HOXA-10 and HOXA-11 and integrin αvβ3 in glandular tissue: low-dose (4 mg/day) vs standard-dose (6 mg/day) estradiol groups.** The data plotted as mean±SD. *P<0.001.

ml in the standard-dose group, with no significant difference (P=0.746; Table 2). At the second visit, estradiol levels were observed to be lower than those recorded at the first visit in both groups, although these changes did not reach statistical significance (P=0.317; Table 2). The mean serum progesterone levels at the first and second visits were not significantly different between the groups (0.16 [0.10, 0.22] ng/ml vs 0.14 [0.06, 0.20] ng/ml, median [IQR]; P=0.215; and 11.69 [7.51, 14.45] ng/ml vs 10.82 [7.59, 13.65] ng/ml; P=0.592, respectively). The mean endometrial thickness on day 12 of the cycle was 7.84±1.69 mm in the low-dose group and 8.75±2.22 mm in the standard-dose group, without a statistically significant difference (P=0.115; Table 2). Pearson's correlation analysis revealed no significant relationship between the mean endometrial thickness and the intensity of immunohistochemical staining for the regulatory proteins HOXA-10 and HOXA-11 or for integrin αvβ3 (r=−0.107, −0.074, and −0.052, respectively; P>0.05).

## Discussion

Achieving a high-quality endometrium is pivotal for successful embryo implantation, especially in in vitro fertilization treatment. Creating a receptive endometrium is crucial for FET cycles, with programmed preparation offering limited monitoring and flexible scheduling [22]. Prior studies indicate that the impact of endometrial thickness on endometrial receptivity is controversial. However, in our practice, we believe that pregnancy rates are highest when endometrial thickness ranges from 8–14 mm in FET cycles [23,24].

This double-blinded, randomized clinical trial assessed the effects of estradiol dosage on artificial endometrial preparation. The results indicated that a standard dose of estradiol (6 mg/day), initiated on day 2 of the menstrual cycle, led to an average endometrial thickness of 8 mm. In contrast, a lower dose (4 mg/day) resulted in slightly less thickening. However, no significant difference in endometrial thickness was observed between the two dosage groups. Histological assessment using the Noyes criteria revealed a similar transformation of the endometrium into the secretory phase across both groups, with adequate secretory-phase characteristics on days 18 and 19 of the cycle. These findings imply that both

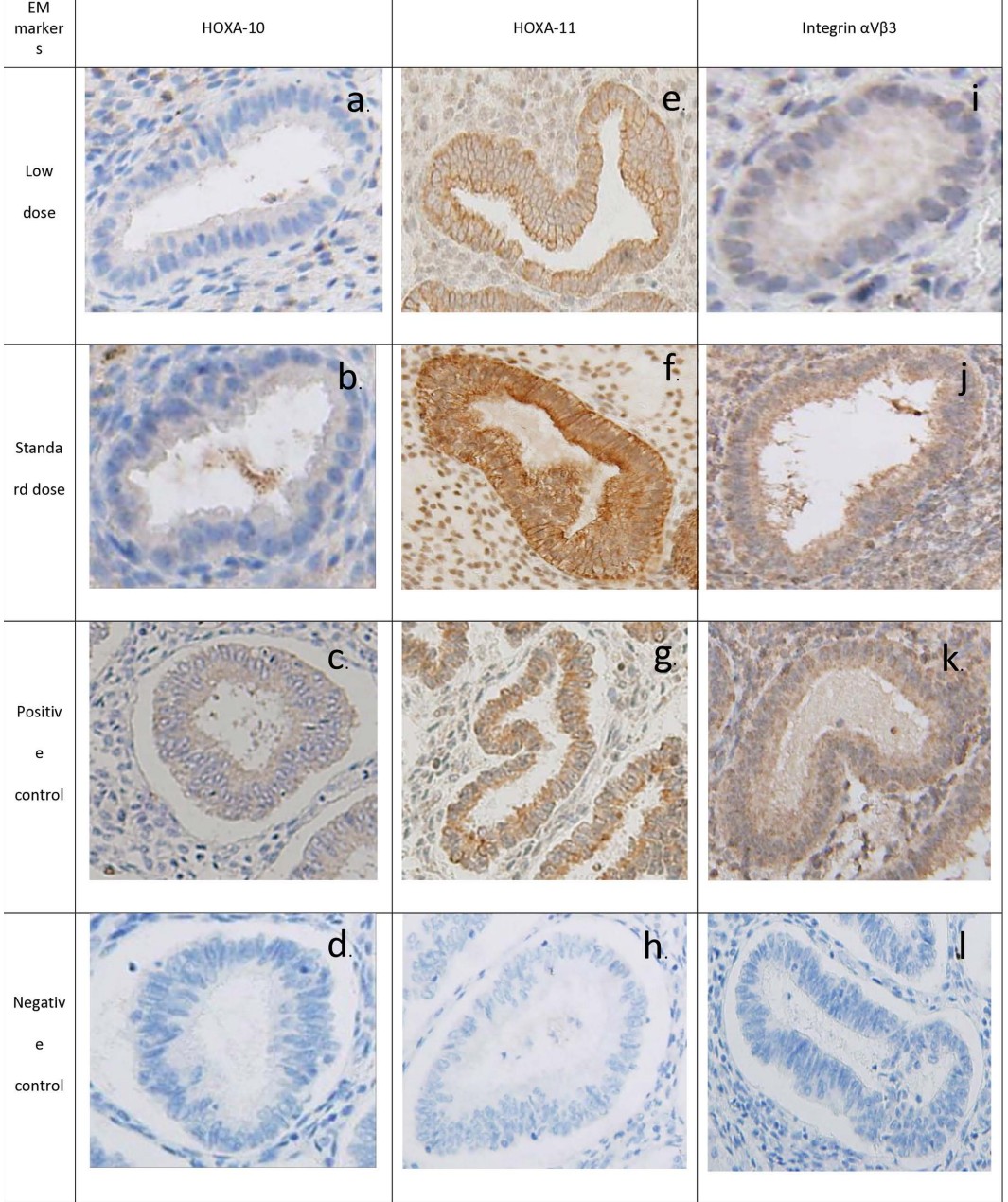

**Fig 3. Differential expression of the regulatory proteins HOXA-10 and HOXA-11 and integrin αvβ3 in endometrial tissue: immunohistochemistry analysis across low estradiol doses (a, e, i) and standard estradiol doses (b, f, J) with positive controls (c, g, k) and negative controls (d, h, l).** Representative images of (a, b, c) HOX A10, (e, f, g) HOX A11, and (i, j, k) integrin αvβ3 expression in low estradiol doses, standard estradiol doses, and positive control. Magnification = x400.

estradiol regimens could be effective for endometrial secretory transformation and initiating artificial endometrial preparation in FET cycles, aligning with the findings of Lewin et al [19].

While assessing mean endometrial thickness serves as a practical morphological marker for endometrial receptivity, it is essential to recognize the influence of various molecular and protein changes on implantation. Lessey et al [25]

**Table 2. Comparative analysis of serum estradiol and progesterone levels and endometrial thickness during artificial cycles among different treatment groups.**

| Variables | Low dose (n = 24) | Standard dose (n = 24) | P |
|---|---|---|---|
| Estradiol on day 12 (pg/ml)[a] | 220.55 ± 117.25 | 229.98 ± 80.06 | 0.746 |
| Estradiol on day 19 (pg/ml)[a] | 181.20 ± 90.71 | 205.72 ± 76.46 | 0.317 |
| Progesterone on day 12 (ng/ml)[b] | 0.16 (0.10, 0.22) | 0.14 (0.06, 0.20) | 0.215 |
| Progesterone on day 19 (ng/ml)[b] | 11.69 (7.51, 14.45) | 10.82 (7.59, 13.65) | 0.592 |
| Endometrial thickness on Day 12 (mm)[a] | 7.84 ± 1.69 | 8.75 ± 2.22 | 0.115 |

Note:[a]The data are presented as means ± standard deviations. Independent *t* test was used to compare between groups.

[b]The data are presented as medians (interquartile ranges). Mann–Whitney U test was used to compare between groups).

previously identified a range of specific proteins and biochemical markers that are indicative of endometrial receptivity during the implantation window. The purpose of the present investigation was to assess the impact of estrogen dosage on the expression of the regulatory proteins HOXA-10 and HOXA-11 and the integrin αvβ3 in endometrial preparations for frozen-thawed embryo transfer. Our findings revealed that, despite the absence of a statistically significant difference in mean endometrial thickness between the standard- and low-dose groups, the expression levels of the regulatory proteins HOXA-10 and HOXA-11 and integrin αvβ3 were notably greater in the standard-dose group. Given the established importance of these biochemical markers in endometrial receptivity, our study suggests that employing a standard dose of estradiol (6 mg/day) in the artificial preparation of the endometrium may enhance its receptivity, potentially improving the outcomes of frozen-thawed embryo transfers.

In this investigation, the cellular localization patterns of HOXA10 and HOXA11 were shown to parallel those observed in prior research [17,18,26]. Indeed, IHC study showed HOXA10 and HOXA11 located abundantly in endometrial grand in low doses but remarkably presented in both the endometrial gland and stomal tissue in standard doses. These proteins are essential for embryo implantation during the receptive phase of the endometrium (the implantation window). These findings underscore the critical role of these genes in the implantation process. Interestingly, the location of HOXA10 and HOXA11 proteins showed in the cytoplasm of glandular epithelial but in the nucleus in stromal cells. The function of these transcription factors in the cytoplasm remained unclear but might be related to the post-transcription process, such as the regulation of protein synthesis. Indeed, the function of HOXA10 and HOXA11 in the endometrial is crucially related to the implantation of embryo. Studies have shown that targeted disruption of HOXA-10 is linked to implantation failure in murine models [27]. In addition, the expression pattern of the integrin αVβ3, recognized as one of the pivotal endometrial receptivity biomarkers associated with infertility [16], was consistent with that documented in earlier research [28]. The localization of the integrin αVβ3, particularly at the apex of luminal and glandular cell surfaces during the window of implantation, has been directly correlated with successful implantation. These findings further emphasize the integral role of the integrin αVβ3 in establishing a receptive endometrial environment conducive to embryo implantation. However, variations in gene expression have been observed during several reproductive anomalies, including endometriosis, polycystic ovarian syndrome (PCOS), leiomyoma, polyps, adenomyosis, and hydrosalpinx [29]

Hormonally, both low and standard estadiol dosages achieved adequate estradiol levels, with no significant difference between groups. The reason why the endometrial thickness and markers of endometrial receptivity are higher in the higher dose group and yet the serum levels are similar is that serum estradiol levels may fluctuate after oral administration. Intriguingly, the serum estradiol concentration decreased following progesterone administration in both groups. This phenomenon may be attributed to progesterone's role in downregulating estrogen receptor concentrations and promoting the metabolic conversion of estradiol to weaker metabolites, thereby influencing the hormonal milieu favorable for implantation [30].

Artificial endometrial preparation, commonly utilized by in vitro fertilization centers, is favored because of its simplicity and manipulability. The mode of progesterone administration for secretory transformation of the endometrium includes oral, intramuscular, and vaginal routes. Although conclusive evidence on the optimal form of progesterone supplementation is lacking [31], the selection between these forms does not significantly influence clinical outcomes such as pregnancy and live birth rates [32]. Patient preferences and convenience are key factors in choosing progesterone preparations. In this study, vaginal micronized progesterone was used, which, although associated with lower systemic serum levels than intramuscular injections, is known for its direct, local absorption, resulting in standarder tissue concentrations in the gynecological tract and endometrium [33]. Baseline serum progesterone levels were established during the first visit, acknowledging that elevated levels in the late follicular phase can markedly alter gene expression in the endometrium [34]. Subsequent serum measurements confirmed progesterone absorption, but no significant intergroup differences were observed.

This study is the first double-blinded, randomized clinical trial to evaluate the impact of different estrogen doses on endometrial receptivity markers during FET cycle preparation. However, the comparison of only two fixed doses of estrogen points to the need for more comprehensive research. Future studies should explore various aspects of endometrial preparation protocols to potentially improve implantation and pregnancy outcomes.

## Conclusions

Identifying the optimal approach for artificial endometrial preparation and the receptive window is crucial for enhancing implantation success in FET cycles. The findings of this study suggest that a fixed daily oral dose of 6 mg of estradiol was associated with improved markers of endometrial receptivity as compared with a fixed daily oral dose of 4 mg of estradiol.

## Supporting information

**S1. Study protocol.**
(DOCX)

**S2. CONSORT Checklist PLOS one.**
(DOCX)

**S3. Data Estogen_endometrium.**
(XLSX)

## Acknowledgments

The authors thank the Pathology Unit staff at Siriraj Hospital for their invaluable assistance with the laboratory procedures and Dr Saowaluck Hunnanggul of the Clinical Epidemiology Unit for her support with the statistical analyses.

## Author contributions

**Conceptualization:** Nichamon Parkpinyo, Suchanan Hanamornroongruang, Pitak Laokirkkiat, Roungsin Choavaratana.

**Data curation:** Nichamon Parkpinyo, Suchanan Hanamornroongruang.

**Formal analysis:** Somsin Petyim, Nichamon Parkpinyo.

**Funding acquisition:** Somsin Petyim.

**Investigation:** Somsin Petyim, Nichamon Parkpinyo, Isarin Thanaboonyawat, Japarath Prechapanich.

**Methodology:** Somsin Petyim, Nichamon Parkpinyo, Suchanan Hanamornroongruang.

**Project administration:** Somsin Petyim, Nichamon Parkpinyo.

**Software:** Nichamon Parkpinyo.

**Supervision:** Somsin Petyim, Pitak Laokirkkiat, Roungsin Choavaratana.

**Validation:** Somsin Petyim.

**Writing – original draft:** Somsin Petyim, Nichamon Parkpinyo, Suchanan Hanamornroongruang.

**Writing – review & editing:** Somsin Petyim, Nichamon Parkpinyo, Isarin Thanaboonyawat.

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
