## [Decision Letter · Decision Letter 0]

PONE-D-24-45040Effects of Standard and Low Doses of Estradiol on Markers of Endometrial Receptivity in Frozen-thawed Embryo Transfer Cycles: Double-blind, Randomized Clinical TrialPLOS ONE

Dear Dr. petyim,

Thank you for submitting your manuscript to PLOS ONE. After careful consideration, we feel that it has merit but does not fully meet PLOS ONE’s publication criteria as it currently stands. Therefore, we invite you to submit a revised version of the manuscript that addresses the points raised during the review process.

We look forward to receiving your revised manuscript.

Kind regards,

Birendra Mishra, DVM, PhD

Academic Editor

PLOS ONE

Journal Requirements:

[Siriraj Grant for Research Development, Faculty of Medicine Siriraj Hospital, Mahidol University, Bangkok, Thailand (grant number R016133010)].

Additional Editor Comments:

In Figure 3, please indicate the positively stained cells using arrows or arrowheads.

Reviewers' comments:

Reviewer's Responses to Questions

**Comments to the Author**

1. Is the manuscript technically sound, and do the data support the conclusions?

Reviewer #1: Partly

Reviewer #2: Yes

Reviewer #3: Yes

Reviewer #4: Partly

2. Has the statistical analysis been performed appropriately and rigorously? 

Reviewer #1: No

Reviewer #2: Yes

Reviewer #3: Yes

Reviewer #4: No

3. Have the authors made all data underlying the findings in their manuscript fully available?

Reviewer #1: Yes

Reviewer #2: Yes

Reviewer #3: Yes

Reviewer #4: Yes

4. Is the manuscript presented in an intelligible fashion and written in standard English?

Reviewer #1: Yes

Reviewer #2: Yes

Reviewer #3: Yes

Reviewer #4: Yes

5. Review Comments to the Author

Reviewer #1: This manuscript reported findings from a double-blind randomized control trial. There were 50 study participants enrolled in the trial but only 48 of them were included in the analysis. There are additional information needed to make this scientific report more complete and rigorous.

1. Information about sample size consideration should be reported in the main context. The attached study protocol does not have such information either.

2. It is not clear if intent-to-treat (ITT) analysis was implemented. There were 2 study participants who did not continue with the intervention and were not included in the final analysis. However, because the study flow chart implied that none of study participants were lost of follow-up, these two study participants should be included in the ITT analysis.

3. It is implied that study compliance was recorded. However, no such information was reported here.

4. There were longitudinal measurements about Estradiol and Progesterone so the corresponding analysis would be more powerful if repeated measurement ANOVA or linear mixed effect models were used for comparison.

5. The statistical analysis section mentioned the use of t-test or Mann–Whitney U. Then it is recommended to specify which test were those p-values reported in each table footnote or figure title calculated from.

Reviewer #2: The authors conducted a well-structured trial to compare endometrial receptivity based on the estrogen intake of patients preparing for frozen-thawed embryo transfer. In addition to evaluating the standard markers used in endometrial preparation protocols—such as estrogen and progesterone levels, as well as endometrial thickness—they also performed an immunohistochemical analysis to assess the expression levels of biomarkers and integrins in the endometrial cell glands. These factors have been previously identified as playing crucial roles in implantation. However, a few suggestions for improvement can be made:

- Although negative Pearson’s correlations were observed, they were not statistically significant and were not included or interpreted in the paper.

- The discussion section should be revised to provide more context, specifically supporting the results with comments on the role and localization of the biomarkers and integrins in the endometrial cells.

- This paragraph could be rewritten for clarity and coherence:

“However, no significant difference in endometrial thickness was observed between the two dosage groups. Histological assessment using the Noyes criteria revealed a similar transformation of the endometrium into the secretory phase across both groups, with adequate secretory-phase characteristics on days 18 and 19 of the cycle. These findings imply that a higher estradiol dose could be more effective for initiating artificial endometrial preparation in FET cycles,”

Reviewer #3: Parkpinyo et al have conducted a clinical study to examine how different doses of estradiol treatment impact markers of the endometrium in patients undergoing in vitro fertilization. While I don't like using quantification of immunohistochemistry images to compare levels of expression, the data are pretty clear that the higher dose of estradiol results in increased expression of Hoxa10, Hoxa11, and Integrin avb3. There are a couple of places where I think some revisions will help the manuscript. At the top of page 13, the sentence "These findings imply that a higher estradiol dose could be more effective for initiating artificial endometrial preparation in FET cycles, aligning with the findings of Lewin et al (19)" comes at the end of a paragraph describing no statistical difference in endometrial thickness or histological assessment. Please revise this sentence as the data discussed do not support a higher dose. On page 11 when you describe the cytoplasmic staining, please discuss how cytoplasmic staining for the transcription factors Hoxa10 and Hoxa11 are having a functional impact on the endometrium.

Reviewer #4: I have reviewed the manuscript with great interest, as it explores differences in endometrial markers between women randomized to low or standard doses of estradiol in the context of frozen embryo transfer (FET) procedures. This is a relevant and timely topic in reproductive medicine, providing valuable insights into the biological effects of hormonal preparation. While the study presents intriguing results, certain methodological and interpretative aspects need further attention to strengthen the manuscript's scientific rigor and clarity.

Sample Size Calculation: The manuscript does not provide a sample size calculation. It is essential to clarify how the sample size was determined to ensure the study is adequately powered.

Objectives and Conclusions: The expression of endometrial markers should not be conflated with clinical outcomes, such as pregnancy or delivery, given the lack of strong evidence linking these markers directly to success rates. I recommend revising the objectives and conclusions to align more closely with the scope of the study, avoiding speculative claims.

Confounding Factors: The authors should discuss potential confounding factors that could influence the expression of the studied markers. For example, what is the expected variability in marker expression? Addressing this would strengthen the validity of the findings.

Presentation of Data in Table 1: Since this is a randomized study, Table 1 should not include p-values. Randomization inherently balances groups, making such comparisons unnecessary and potentially misleading.

6. PLOS authors have the option to publish the peer review history of their article (what does this mean? ). If published, this will include your full peer review and any attached files.

**Do you want your identity to be public for this peer review?** For information about this choice, including consent withdrawal, please see our Privacy Policy .

Reviewer #1: No

Reviewer #2: No

Reviewer #3: No

Reviewer #4: No

---

## [Author Response · Author response to Decision Letter 1]

31 Jan 2025

Response to reviewers

Review Comments to the Author

Reviewer #1: This manuscript reported findings from a double-blind randomized control trial. There were 50 study participants enrolled in the trial but only 48 of them were included in the analysis. There are additional information needed to make this scientific report more complete and rigorous.

1.1. Information about sample size consideration should be reported in the main context. The attached study protocol does not have such information either.

Answer: This information has been stated in line 208-211 that in the low-dose group, one patient’s data was excluded due to a diagnosis of endometrioid adenocarcinoma of the uterus identified in the final pathological report. In the standard-dose group, data from one patient was discarded due to incorrect drug consumption. Therefore, the analyses were performed on a per-protocol basis. Sample size calculation has been added (line 202-205)

1.2. It is not clear if intent-to-treat (ITT) analysis was implemented. There were 2 study participants who did not continue with the intervention and were not included in the final analysis. However, because the study flow chart implied that none of the study participants were lost of follow-up, these two study participants should be included in the ITT analysis.

Answer: This study, we intended to perform per-protocol analysis. That’s why we didn’t include 2 study participants who did not continue with the intervention (mentioned in line 208-211).

1.3. It is implied that study compliance was recorded. However, no such information was reported here.

Answer: Study compliance was mentioned in line 145-146.

1.4. There were longitudinal measurements about Estradiol and Progesterone so the corresponding analysis would be more powerful if repeated measurement ANOVA or linear mixed effect models were used for comparison.

Answer: Thank you for your suggestions.

1.5. The statistical analysis section mentioned the use of t-test or Mann–Whitney U. Then it is recommended to specify which test were those p-values reported in each table footnote or figure title calculated from.

Answer: The statistical analysis was mentioned in the table footnote (Table 2).

Reviewer #2: The authors conducted a well-structured trial to compare endometrial receptivity based on the estrogen intake of patients preparing for frozen-thawed embryo transfer. In addition to evaluating the standard markers used in endometrial preparation protocols—such as estrogen and progesterone levels, as well as endometrial thickness—they also performed an immunohistochemical analysis to assess the expression levels of biomarkers and integrins in the endometrial cell glands. These factors have been previously identified as playing crucial roles in implantation. However, a few suggestions for improvement can be made:

2.1. Although negative Pearson’s correlations were observed, they were not statistically significant and were not included or interpreted in the paper.

Answer: Thank you for your suggestions.

2.2. The discussion section should be revised to provide more context, specifically supporting the results with comments on the role and localization of the biomarkers and integrins in the endometrial cells.

Answer: Thank you for your suggestions. The discussion part was revised, The role and localization of the HOXA and integrins were mentioned in the 4th paragraph of discussion part (line 284-289)

2.3. This paragraph could be rewritten for clarity and coherence:

“However, no significant difference in endometrial thickness was observed between the two dosage groups. Histological assessment using the Noyes criteria revealed a similar transformation of the endometrium into the secretory phase across both groups, with adequate secretory-phase characteristics on days 18 and 19 of the cycle. These findings imply that a higher estradiol dose could be more effective for initiating artificial endometrial preparation in FET cycles,”

Answer: Thank you for your suggestions. This paragraph has already been rewritten (line261-263).

Reviewer #3: Parkpinyo et al have conducted a clinical study to examine how different doses of estradiol treatment impact markers of the endometrium in patients undergoing in vitro fertilization. While I don't like using quantification of immunohistochemistry images to compare levels of expression, the data are pretty clear that the higher dose of estradiol results in increased expression of Hoxa10, Hoxa11, and Integrin avb3. There are a couple of places where I think some revisions will help the manuscript.

3.1. At the top of page 13, the sentence "These findings imply that a higher estradiol dose could be more effective for initiating artificial endometrial preparation in FET cycles, aligning with the findings of Lewin et al (19)" comes at the end of a paragraph describing no statistical difference in endometrial thickness or histological assessment. Please revise this sentence as the data discussed do not support a higher dose.

Answer: Thank you for your suggestions. This paragraph has already been rewritten (line 261-263).

3.2. On page 11 when you describe the cytoplasmic staining, please discuss how cytoplasmic staining for the transcription factors Hoxa10 and Hoxa11 are having a functional impact on the endometrium.

Answer: Thank you for your suggestions. The role and localization of the HOXA and integrins were mentioned in the 4th paragraph of discussion part (line 277-289)

Reviewer #4: I have reviewed the manuscript with great interest, as it explores differences in endometrial markers between women randomized to low or standard doses of estradiol in the context of frozen embryo transfer (FET) procedures. This is a relevant and timely topic in reproductive medicine, providing valuable insights into the biological effects of hormonal preparation. While the study presents intriguing results, certain methodological and interpretative aspects need further attention to strengthen the manuscript's scientific rigor and clarity.

4.1. Sample Size Calculation: The manuscript does not provide a sample size calculation. It is essential to clarify how the sample size was determined to ensure the study is adequately powered.

Answer: Thank you for your suggestion. Sample size calculation has been added (line 202-205)

In brief: Sample size calculation

Testing two independent means formula was used. The study from Yang Y. et al was used for sample size calculation. a = Type I error = 0.05, 1-β = Power = 0.9, β = 0.1, µ1 = 218, µ 2 = 263, ơ1 = 46, ơ 2 = 47. Testing two independent means formula:

The sample sizes are 46 women. When dropout rate = 10 %, Total number of sample size are 50 women.

4.2. Objectives and Conclusions: The expression of endometrial markers should not be conflated with clinical outcomes, such as pregnancy or delivery, given the lack of strong evidence linking these markers directly to success rates. I recommend revising the objectives and conclusions to align more closely with the scope of the study, avoiding speculative claims.

Answer: Thank you for your suggestion. The objective of this study aimed to evaluate the impact of two different estrogen doses on endometrial receptivity markers during the window of implantation in endometrial preparation for frozen-thawed embryo transfer. The conclusion is that a fixed daily oral dose of 6mg of estradiol was associated with improved markers of endometrial receptivity as compared with a fixed daily oral dose of 4 mg of estradiol.

4.3. Confounding Factors: The authors should discuss potential confounding factors that could influence the expression of the studied markers. For example, what is the expected variability in marker expression? Addressing this would strengthen the validity of the findings.

Answer: Thank you for your suggestion. Potential confounding factors have been added in line 287-289.

4.4. Presentation of Data in Table 1: Since this is a randomized study, Table 1 should not include p-values. Randomization inherently balances groups, making such comparisons unnecessary and potentially misleading.

Answer: P-value has been removed from Table1.

---

## [Decision Letter · Decision Letter 1]

PONE-D-24-45040R1Effects of Standard and Low Doses of Estradiol on Markers of Endometrial Receptivity in Frozen-thawed Embryo Transfer Cycles: Double-blind, Randomized Clinical TrialPLOS ONE

Dear Dr.  petyim,

Thank you for submitting your manuscript to PLOS ONE. After careful consideration, we feel that it has merit but does not fully meet PLOS ONE’s publication criteria as it currently stands. Therefore, we invite you to submit a revised version of the manuscript that addresses the points raised during the review process.

We look forward to receiving your revised manuscript.

Kind regards,

Ayman A Swelum

Academic Editor

PLOS ONE

Journal Requirements:

Additional Editor Comments:

**The manuscript was improved. however, some points must be addressed before its acceptance.**

Reviewers' comments:

Reviewer's Responses to Questions

**Comments to the Author**

1. If the authors have adequately addressed your comments raised in a previous round of review and you feel that this manuscript is now acceptable for publication, you may indicate that here to bypass the “Comments to the Author” section, enter your conflict of interest statement in the “Confidential to Editor” section, and submit your "Accept" recommendation.

Reviewer #1: All comments have been addressed

Reviewer #2: All comments have been addressed

Reviewer #3: (No Response)

2. Is the manuscript technically sound, and do the data support the conclusions?

Reviewer #1: Partly

Reviewer #2: Yes

Reviewer #3: Yes

3. Has the statistical analysis been performed appropriately and rigorously? 

Reviewer #1: No

Reviewer #2: Yes

Reviewer #3: Yes

4. Have the authors made all data underlying the findings in their manuscript fully available?

Reviewer #1: Yes

Reviewer #2: Yes

Reviewer #3: Yes

5. Is the manuscript presented in an intelligible fashion and written in standard English?

Reviewer #1: Yes

Reviewer #2: Yes

Reviewer #3: Yes

6. Review Comments to the Author

Reviewer #1: 1. Authors have added in sample size justification (lines 202-205). However, key estimates used in such sample size calculation such as effect size and corresponding statistical test are missing. In addition, Lines 109-110 said that the primary outcome is the expression levels of three genes. So multiple testing adjustment should be considered in the sample size calculation.

2. There is still no report about study compliance. Lines 145-146 described how study compliance was recorded. Were all 48 patients in the final analysis compliant to study protocol?

3. All p-values in Table 1 have been removed but Line 214 still added a p-value for comparing age between two dose groups.

Reviewer #2: (No Response)

Reviewer #3: I apologize for lack of specificity in my comment to be address. My concern is that Hoxa10 and Hoxa11 are observed in the cytoplasm, which is an unusual location for a transcription factor that may have a functional role in the endometrium. The references you site for expression of these genes in the endometrium (17, 18, and 26) report both cytoplasmic and nuclear localization. Can you could comment on the functional significance for Hoxa10 and Hoxa11 being localized in the cytoplasm?

7. PLOS authors have the option to publish the peer review history of their article (what does this mean? ). If published, this will include your full peer review and any attached files.

**Do you want your identity to be public for this peer review?** For information about this choice, including consent withdrawal, please see our Privacy Policy .

Reviewer #1: No

Reviewer #2: No

Reviewer #3: No

---

## [Author Response · Author response to Decision Letter 2]

7 May 2025

Response to Reviewer Comments

Reviewer #1:

1. Authors have added in sample size justification (lines 202-205). However, key estimates used in such sample size calculation such as effect size and corresponding statistical test are missing. In addition, Lines 109-110 said that the primary outcome is the expression levels of three genes. So multiple testing adjustment should be considered in the sample size calculation.

Answer: Thank you for your suggestion. In the study, the appropriate statistical test is the Independent samples t-test. The effect size is measured using Cohen’s d: (Mentioned in line 205-206)

Moreover, multiple testing adjustment was actually considered in the sample size calculation and sample size for HOXA-10 was the greatest number. So, authors used this sample size as a reference number for sample size calculation.

2. There is still no report about study compliance. Lines 145-146 described how study compliance was recorded. Were all 48 patients in the final analysis compliant to study protocol?

Answer: Thank you for your comment. Indeed, all 48 patients in the final analysis were compliant to study protocol. But in the low-dose group, one patient’s data were excluded due to a diagnosis of endometrioid adenocarcinoma of the uterus identified in the final pathological report. In the standard-dose group, data from one patient were discarded due to incorrect drug consumption. Therefore, the analyses were performed on a per-protocol basis (24 patients in each group) (Mentioned in line 209-213)

3. All p-values in Table 1 have been removed but Line 214 still added a p-value for comparing age between two dose groups.

Answer: Thank you for your comment. P-value were deleted in line 216-217.

Reviewer #2: (No Response)

Reviewer #3: I apologize for lack of specificity in my comment to be address. My concern is that Hoxa10 and Hoxa11 are observed in the cytoplasm, which is an unusual location for a transcription factor that may have a functional role in the endometrium. The references you site for expression of these genes in the endometrium (17, 18, and 26) report both cytoplasmic and nuclear localization. Can you could comment on the functional significance for Hoxa10 and Hoxa11 being localized in the cytoplasm?

Answer: Thank you for your comment. I agree that Hoxa10 and Hoxa11 are transcription factors mainly located in the nucleus. However, the result in the study showed the expression of Hoxa10 demonstrated mainly in the cytoplasm of the glandular epithelial area, the Hoxa10 protein is localized in the nucleus of stroma cells. These findings are similar to the study from Yang 2017 (17). The function of these transcription factors in the cytoplasm, however, remained unclear but might be related to the post-transcription process, such as the regulation of protein synthesis. We added in the discussion in line 286-292.

---

## [Decision Letter · Decision Letter 2]

PONE-D-24-45040R2Effects of Standard and Low Doses of Estradiol on Markers of Endometrial Receptivity in Frozen-thawed Embryo Transfer Cycles: Double-blind, Randomized Clinical TrialPLOS ONE

Dear Dr. Petyim,

Thank you for submitting your manuscript to PLOS ONE. After careful consideration, we feel that it has merit but does not fully meet PLOS ONE’s publication criteria as it currently stands. Therefore, we invite you to submit a revised version of the manuscript that addresses the points raised during the review process.

**ACADEMIC EDITOR: Please respond carefully to the reviewer comments ** . 

We look forward to receiving your revised manuscript.

Kind regards,

Ayman A Swelum

Academic Editor

PLOS ONE

Journal Requirements:

Reviewers' comments:

Reviewer's Responses to Questions

**Comments to the Author**

1. If the authors have adequately addressed your comments raised in a previous round of review and you feel that this manuscript is now acceptable for publication, you may indicate that here to bypass the “Comments to the Author” section, enter your conflict of interest statement in the “Confidential to Editor” section, and submit your "Accept" recommendation.

Reviewer #1: All comments have been addressed

Reviewer #3: All comments have been addressed

2. Is the manuscript technically sound, and do the data support the conclusions?

Reviewer #1: Partly

Reviewer #3: Yes

3. Has the statistical analysis been performed appropriately and rigorously? 

Reviewer #1: No

Reviewer #3: Yes

4. Have the authors made all data underlying the findings in their manuscript fully available?

Reviewer #1: Yes

Reviewer #3: Yes

5. Is the manuscript presented in an intelligible fashion and written in standard English?

Reviewer #1: Yes

Reviewer #3: Yes

6. Review Comments to the Author

Reviewer #1: It appears that the authors are still unclear about the application of multiple testing adjustment. The current version of the manuscript states that a type I error rate of 0.05 was used for the sample size calculation, which does not indicate that any correction for multiple comparisons was applied. Additionally, the authors mention that the sample size was based on HOXA-10, the gene requiring the largest sample, suggesting that different effect sizes were assumed for the three genes. These assumptions, along with the rationale for using different effect sizes across genes, should be explicitly described and justified in the main text.

Reviewer #3: (No Response)

7. PLOS authors have the option to publish the peer review history of their article (what does this mean? ). If published, this will include your full peer review and any attached files.

**Do you want your identity to be public for this peer review?** For information about this choice, including consent withdrawal, please see our Privacy Policy .

Reviewer #1: No

Reviewer #3: No

---

## [Author Response · Author response to Decision Letter 3]

13 Jun 2025

Response to Reviewer Comments (13062025)

Reviewer #1: It appears that the authors are still unclear about the application of multiple testing adjustment. The current version of the manuscript states that a type I error rate of 0.05 was used for the sample size calculation, which does not indicate that any correction for multiple comparisons was applied. Additionally, the authors mention that the sample size was based on HOXA-10, the gene requiring the largest sample, suggesting that different effect sizes were assumed for the three genes. These assumptions, along with the rationale for using different effect sizes across genes, should be explicitly described and justified in the main text.

Answer: Thank you for your comment. I agree with you.

The current version of the manuscript states that a type I error rate of 0.05 was used for the sample size calculation. This is because we defined one primary outcome (HOXA-10) and treat the others as secondary for calculation. So, we used alpha = 0.05 for that one outcome in sample size calculation. We’ve already revised the primary and secondary outcomes as in line 54-56 & 109-113.

Additionally, the sample size was calculated based on HOXA-10, as this gene demonstrated the largest effect size and required the greatest number of participants. HOXA-10 functions as an upstream master regulator, controlling the expression of other key genes involved in uterine development and endometrial receptivity (mentioned in line 203-206).

---

## [Decision Letter · Decision Letter 3]

PONE-D-24-45040R3Effects of Standard and Low Doses of Estradiol on Markers of Endometrial Receptivity in Frozen-thawed Embryo Transfer Cycles: Double-blind, Randomized Clinical TrialPLOS ONE

Dear Dr. Petym,

Thank you for submitting your manuscript to PLOS ONE. After careful consideration, we feel that it has merit but does not fully meet PLOS ONE’s publication criteria as it currently stands. Therefore, we invite you to submit a revised version of the manuscript that addresses the points raised during the review process.

**ACADEMIC EDITOR: Please respond carefully to the reviewer comment.**

We look forward to receiving your revised manuscript.

Kind regards,

Ayman A Swelum

Academic Editor

PLOS ONE

Journal Requirements:

Reviewers' comments:

Reviewer's Responses to Questions

**Comments to the Author**

1. If the authors have adequately addressed your comments raised in a previous round of review and you feel that this manuscript is now acceptable for publication, you may indicate that here to bypass the “Comments to the Author” section, enter your conflict of interest statement in the “Confidential to Editor” section, and submit your "Accept" recommendation.

Reviewer #1: All comments have been addressed

2. Is the manuscript technically sound, and do the data support the conclusions?

Reviewer #1: Partly

3. Has the statistical analysis been performed appropriately and rigorously? 

Reviewer #1: No

4. Have the authors made all data underlying the findings in their manuscript fully available?

Reviewer #1: Yes

5. Is the manuscript presented in an intelligible fashion and written in standard English?

Reviewer #1: Yes

6. Review Comments to the Author

Reviewer #1: It is now clear to readers that HOXA-10 is the sole gene selected as the primary outcome. However, the justification provided for its use in the sample size calculation—namely, that "this gene demonstrated the largest effect size and required the greatest number of participants" (lines 204–205)—is incorrect. In fact, detecting a larger effect size typically requires fewer, not more, participants. This rationale should be corrected before the manuscript can be accepted.

7. PLOS authors have the option to publish the peer review history of their article (what does this mean? ). If published, this will include your full peer review and any attached files.

**Do you want your identity to be public for this peer review?** For information about this choice, including consent withdrawal, please see our Privacy Policy .

Reviewer #1: No

---

## [Author Response · Author response to Decision Letter 4]

30 Jun 2025

Answer: Thank you for your comment. Your comment is very useful and I totally agree with you. I have corrected as your comment.

The sample size calculation was based on the previous study (17). The sample size was calculated based on HOXA-10. HOXA-10 functions as an upstream master regulator, controlling the expression of other key genes involved in uterine development and endometrial receptivity (17).

(mentioned in line 203-205).

---

## [Decision Letter · Decision Letter 4]

Effects of Standard and Low Doses of Estradiol on Markers of Endometrial Receptivity in Frozen-thawed Embryo Transfer Cycles: Double-blind, Randomized Clinical Trial

PONE-D-24-45040R4

Dear Dr. petyim,

We’re pleased to inform you that your manuscript has been judged scientifically suitable for publication and will be formally accepted for publication once it meets all outstanding technical requirements.

Kind regards,

Ayman A Swelum

Academic Editor

PLOS ONE

Additional Editor Comments (optional):

Reviewers' comments:

Reviewer's Responses to Questions

**Comments to the Author**

1. If the authors have adequately addressed your comments raised in a previous round of review and you feel that this manuscript is now acceptable for publication, you may indicate that here to bypass the “Comments to the Author” section, enter your conflict of interest statement in the “Confidential to Editor” section, and submit your "Accept" recommendation.

Reviewer #1: All comments have been addressed

2. Is the manuscript technically sound, and do the data support the conclusions?

Reviewer #1: Yes

3. Has the statistical analysis been performed appropriately and rigorously? 

Reviewer #1: Yes

4. Have the authors made all data underlying the findings in their manuscript fully available?

Reviewer #1: Yes

5. Is the manuscript presented in an intelligible fashion and written in standard English?

Reviewer #1: Yes

6. Review Comments to the Author

Reviewer #1: The manuscript has been corrected and now reach the level for publication. All my concerns are addressed.

7. PLOS authors have the option to publish the peer review history of their article (what does this mean? ). If published, this will include your full peer review and any attached files.

**Do you want your identity to be public for this peer review?** For information about this choice, including consent withdrawal, please see our Privacy Policy .

Reviewer #1: No

---

## [Editor Report · Acceptance letter]

PONE-D-24-45040R4

PLOS ONE

Dear Dr. Petyim,

I'm pleased to inform you that your manuscript has been deemed suitable for publication in PLOS ONE. Congratulations! Your manuscript is now being handed over to our production team.

Kind regards,

on behalf of

Professor Ayman A Swelum

Academic Editor

PLOS ONE